# Selenium single-wavelength anomalous diffraction *de novo* phasing using an X-ray-free electron laser

Mark S. Hunter[1], Chun Hong Yoon[1], Hasan DeMirci[2,3,4], Raymond G. Sierra[1,2], E. Han Dao[2,3], Radman Ahmadi[1], Fulya Aksit[2], Andrew L. Aquila[1], Halilibrahim Ciftci[1], Serge Guillet[1], Matt J. Hayes[1], Thomas J. Lane[1,3], Meng Liang[1], Ulf Lundström[3], Jason E. Koglin[1], Paul Mgbam[1], Yashas Rao[1], Lindsey Zhang[1], Soichi Wakatsuki[1,3], James M. Holton[4,5] & Sébastien Boutet[1]

Structural information about biological macromolecules near the atomic scale provides important insight into the functions of these molecules. To date, X-ray crystallography has been the predominant method used for macromolecular structure determination. However, challenges exist when solving structures with X-rays, including the phase problem and radiation damage. X-ray-free electron lasers (X-ray FELs) have enabled collection of diffraction information before the onset of radiation damage, yet the majority of structures solved at X-ray FELs have been phased using external information via molecular replacement. *De novo* phasing at X-ray FELs has proven challenging due in part to per-pulse variations in intensity and wavelength. Here we report the solution of a selenobiotinyl-streptavidin structure using phases obtained by the anomalous diffraction of selenium measured at a single wavelength (Se-SAD) at the Linac Coherent Light Source. Our results demonstrate Se-SAD, routinely employed at synchrotrons for novel structure determination, is now possible at X-ray FELs.

[1] Linac Coherent Light Source, SLAC National Accelerator Laboratory, Menlo Park, California 94025, USA. [2] Stanford PULSE Institute, SLAC National Accelerator Laboratory, Menlo Park, California 94025, USA. [3] Biosciences Division, SLAC National Accelerator Laboratory, Menlo Park, California 94025, USA. [4] Stanford Synchrotron Radiation Lightsource, SLAC National Accelerator Laboratory, Menlo Park, California 94025, USA. [5] Department of Biochemistry and Biophysics, University of California, San Francisco, California 94158, USA. Correspondence and requests for materials should be addressed to M.S.H. (email: mhunter2@slac.stanford.edu).

Despite its widespread use within structural biology, X-ray crystallography entails several fundamental challenges, including the crystallographic phase problem and X-ray induced radiation damage. The phase problem arises from the need to know both the amplitudes and the phases of the scattered X-rays in order to reconstruct the electron density of the sample. Measurements made in X-ray crystallography only yield the amplitude information; the phase information is lost and must be derived computationally or experimentally. One way around this phase problem is to assume phases of a closely related molecule, a method known as molecular replacement[1]. However, for truly novel structures, the molecular replacement method does not suffice and *de novo* phasing becomes necessary. Experimental methods towards this end include single/multiple isomorphous replacement[2] and single/multiple-wavelength anomalous diffraction (SAD/MAD)[3].

Of these experimental phasing techniques, SAD phasing has become the most prevalent method for *de novo* protein structure determination, accounting for over 70% of *de novo* structures deposited to the Protein Data Bank (www.wwpdb.org) in 2013 (ref. 4). The replacement of methionine residues with seleno-methionine[5] residues has allowed generalized *de novo* macromolecular structure determination using SAD based upon the selenium K-edge (Se-SAD), in a method colloquially referred to as 'the magic bullet' of structural biology[6], as methionine is present in virtually all proteins. Further, Se-SAD experiments have been greatly facilitated by synchrotron X-ray facilities, which offer highly tunable X-ray energies and high X-ray intensities. However, as with all measurements at synchrotrons, care must be taken to avoid X-ray-induced radiation damage, which makes measurements at room temperature and using microcrystals challenging.

X-ray crystallography is an intrinsically destructive method, causing damage at the molecular level by depositing energy throughout the sample via photo-absorption[7]. Therefore, a careful balance must always be found to allow the structures determined to have minimal X-ray induced radiation damage while still collecting high-resolution data. Large, well-ordered crystals will generally allow high-resolution data to be collected while minimizing radiation damage but many biologically interesting macromolecules, for example, membrane proteins and large macromolecule complexes, are difficult to crystallize and often only yield small crystals. Consequently, the two goals of circumventing radiation damage while achieving high-resolution structural information have generally opposed each other. New X-ray sources, known as X-ray-free electron lasers (X-ray FELs), are fundamentally altering the relationship between X-ray induced radiation damage and the structural resolution achieved.

X-ray FELs, such as the Linac Coherent Light Source (LCLS), produce extremely intense X-ray pulses with very short pulse durations. The femtosecond pulses of X-ray FELs have been shown in theory[8] and in practice[9,10] to permit the collection of structural information from crystals before the onset of significant damage. The method of serial femtosecond crystallography (SFX) has pushed the frontiers of protein crystallography at X-ray FELs through the use of smaller crystals than typically used with conventional sources[10–12] and the use of the unique time resolution afforded by X-ray FELs[13–15].

The structures solved to date using SFX have mainly been determined using phases assumed through molecular replacement. For SFX to contribute novel structural information, routine *de novo* phasing from X-ray FEL data is necessary. The first indication that SAD/MAD phasing might be feasible at an X-ray FEL was the detection of an anomalous signal for sulfur in a lysozyme data set collected by SFX[16]. Other preliminary successes with *de novo* structure determination at

X-ray FELs have since been reported, including successfully phasing a lysozyme structure using the anomalous signal from gadolinium[17], and successfully phasing luciferin-regenerating enzyme using a combination of isomorphous replacement and anomalous diffraction data from mercury[18]. Recently, sulfur phasing was achieved at (SPring-8 Angstrom Compact free electron Laser) SACLA in Japan[19] and LCLS[20]. Yet, *de novo* phasing with SFX data remains a challenging endeavor. Extending the Se-SAD phasing 'magic bullet' of structural biology to SFX experiments promises to provide a routine method for *de novo* phasing at X-rays FELs.

Here we show the successful use of single-wavelength anomalous diffraction to solve the crystal structure of streptavidin bound to a selenobiotin molecule to 1.9 Å. LCLS was operated at energies not previously available, delivering the 12.8 keV X-rays necessary for the work. The structure was solved using weak anomalous difference data indicating a general method for exploring the parameter space associated with *de novo* phasing that could be applied to other difficult systems or weak data. This demonstration shows that Se-SAD is now possible at LCLS and will be available as a non-standard configuration for the community.

## Results

**Experimental set-up**. To determine the feasibility of Se-SAD phasing at LCLS using SFX, we collected a highly redundant data set to 1.9 Å of selenobiotinyl-streptavidin (Se-B SA) co-crystals ($10 \times 10 \times 2 \, \mu m$), in which the sulfur atom of the biotin molecule is replaced by selenium[3]. The experiments were performed using a serial operation of the serial crystallography set-up at the Coherent X-ray Imaging instrument (CXI) of LCLS in a manner similar to a previous report[21]: the unused beam from an upstream experiment was refocused for a downstream experiment, allowing simultaneous data collection from two separate sample chambers. Individual X-ray pulses of nominally 12.8 keV photon energy and 0.93 mJ average pulse energy with pulse duration between 35 and 40 fs were focused into a spot size of approximately $3 \times 3 \, \mu m$ in both the upstream and downstream interaction regions using beryllium compound refractive lenses. Microcrystals of the Se-B SA complex were delivered to the X-ray interaction region using a concentric microfluidic electrokinetic sample holder (coMESH)[22] operating at nominally $1 \, \mu l \, min^{-1}$ in each chamber; two identical, independent coMESH systems were simultaneously operated during data collection in a similar experimental set-up to that described previously[21]. Data for the two concurrent experiments were recorded on separate, 2.3 Megapixel Cornell-SLAC Pixel Array Detectors[23] at 120 Hz operation each. Less than 15 mg of sample was consumed during the 18 h of combined data collection (that is, 9 h per experiment).

**Data collection and data quality**. Diffraction data were collected with pulse energies of 0.093, 0.46 and 0.93 mJ, with the 0.093 mJ pulse energy used to ensure accurate measurements of the low-resolution reflections. During data analysis, the data from both chambers and all pulse energies/transmissions were combined into the final data set, with saturated peaks being rejected from the integration process. A total of 1,567,793 diffraction patterns were identified as potential crystal hits using the *CHEETAH* software[24] (Supplementary Fig. 1). *CrystFEL*[25] parameters (required for peak finding and indexing) were optimized using brute-force grid search and subsequently the unit cell was determined. There were 559,194 (36%) indexed patterns, where patterns with high median background at low scattering angles were subsequently rejected. The resulting 481,079 (31%) patterns were then merged with process_hkl by

only considering unsaturated peaks and reflections with more than seven partial measurements, followed by intensity scaling. Unfortunately, phasing success could not be compared within the X-ray transmission series data subsets, since the data from any subset were insufficient for successful phasing.

For the large data sets acquired in SFX, the metric $R_{split}$ is used to test the internal consistency of the data[25]. It is calculated by dividing the data set into two halves, integrating the two halves separately to obtain two sets of intensities, and comparing them (Supplementary Note 1). Using a resolution cutoff at 1.9 Å, an $R_{split}$ of 4.8 (39.5)% was obtained along with a $CC_{1/2} = 0.998$ (0.762) over the entire (highest) resolution range, demonstrating good internal agreement. The resolution cutoff was chosen to ensure a $CC_{1/2}$ value $> 0.2$ in the final resolution shell. The overall $CC_{ano}$ of the data was 0.177.

**Broad search for solutions**. The structure was solved with a combination of the *PHENIX*[26] and *CCP4* suites[27] using a combinatorial grid search of density modification parameters. The SFX structure was solved by first converting the output MTZ reflection file from CrystFEL-0.6.2 (ref. 25) into the CCP4 suite[27] using the programs *POINTLESS*, *AIMLESS* and *TRUNCATE*[28]. This was necessary to place the Friedel opposite reflections into the conventional asymmetric unit as well as to adjust the estimated standard deviations; an error scale factor between 0.4 and 0.9 was necessary for subsequent phasing calculations to succeed. 'Success' was defined as the eventual discovery of at least one non-crystallographic symmetry (NCS) operator in the density-modified map by *phenix.find_ncs*[26]. The four selenium sites were found using *phenix.hyss*[26], and heavy-atom phase probabilities computed by one cycle of the *CCP4* program *MLPHARE*. Any cycles of refinement in *MLPHARE* were found to degrade the phasing signal. The Hendrickson-Lattman coefficients[29] obtained were then scaled up by a factor of 3–6 using the *CCP4* program *SFTOOLS*. This 'sharpens' the phases, or reduces the estimate of phase error supplied to the density modification step. This phase sharpening increased the final success rate by 20-fold over the unsharpened phases. Density modification was carried out using the *CCP4* program *DM*, with the majority of successes found after running 300–700 cycles of solvent flattening and histogram matching with the default automatic weighting scheme. After this first round of density modification, phenix.find_ncs was run to search for a (NCS) operator. The success of NCS operator discovery was found to be a powerful indicator of downstream phase quality, and served as an excellent filter for error scaling and phase sharpening parameters. Once discovered, the NCS operator was applied for a second round of density modification, starting over with the same sharpened *MLPHARE* phases, but this time adding NCS averaging with a spherical mask centered on the 'guessed

molecular center.' Phases of sufficient quality for automatic model building were obtained with spherical averaging mask radius ranging from 5 to 40 Å and auto-updating the mask every 2–50 cycles of *DM*. Starting with an automatically-derived mask never produced useful phases. In total, over 200,000 *DM* trials were run, and when these trials were filtered for those where an NCS operator was found and subsequently sorted on the figure of merit from density modification, the trial with the highest figure of merit corresponded to phases of sufficient quality to automatically build the structure. Therefore, although this phasing strategy is not the default in any extant automation package, it does represent a robust phasing strategy that leads to success when the trials are scored by readily available metrics. Supplementary Fig. 2 shows a plot of the correlation of the final maps with the known structure and anomalous peak/root-mean-square (RMS)[30] ratio as a function of the percentage of the total data used.

Of the 159,101 jobs run, 178 reported a final figure of merit $> 0.88$, all of which produced high quality maps suitable for automated model building (Fig. 1). The success of NCS operator discovery was found to be a powerful indicator of downstream phase quality, and served as an excellent filter for error scaling and phase sharpening parameters. The final structure produced an $R_{work} = 16.6\%$ and $R_{free} = 19.9\%$ and showed the presence of strong Se peaks, as shown in Fig. 2 and as a stereo image in Supplementary Fig. 3.

**Structure refinement**. Crystallographic refinement of the model was performed with the program *PHENIX*[26]. All the models were checked and completed with *COOT*[31]. Final model of structure contain residues 16–134 in chain A, B, C and D respectively. Residues (48–50) of chain D were disordered and not included in the final structure. The stereochemical quality of the structure was assessed with *PROCHECK*[32]. The Ramachandran statistics (most favoured/additionally allowed/generously allowed) are 89.3/10.5/0.2%. The refinement statistics are summarized in Table 1. All X-ray crystal-structure figures were produced with *PyMOL* (http://www.schrodinger.com/pymol).

**Discussion**
The Se-B SA co-crystals proved to be a challenging system with which to attempt Se-SAD for the first time using SFX at LCLS. It has a low ratio of selenium per asymmetric unit (one selenium atom per 128 amino acids) and low lattice symmetry (P2₁), meaning the measureable anomalous difference signal was quite small. The introduction of the sample using the coMESH injector allowed for relatively low sample consumption ($<15$ mg of protein was used during 18 h of combined data collection time). The coMESH was compatible with the moderately viscous

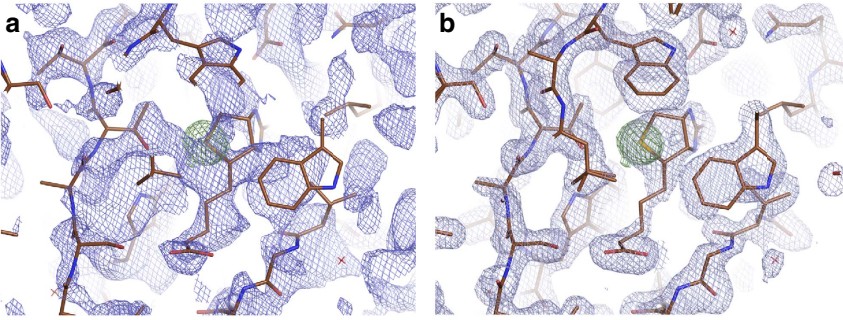

**Figure 1 | Electron density at selenobiotin site.** (**a**) Electron density of the selenobiotin site before solvent flattening and (**b**) Electron density of the final, refined structure. Model from known structure is shown in both frames and green electron density indicates the location of the Se atom.

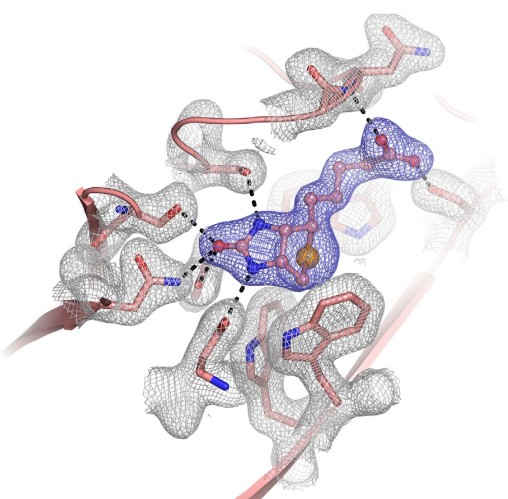

**Figure 2 | Electron-density map of the selenobiotin-binding site.** Final *2Fo-Fc* electron-density map of selenobiotin-binding pocket contoured at the 1σ level. Selenobiotin is shown in balls and sticks model and its electron-density coloured in blue. The locations of active site residues are shown and their electron-density is coloured in grey.

| | Streptavidin selenobiotin (XFEL) |
|---|---|
| **Table 1 | Data collection and refinement statistics.** | |
| *Data collection* | |
| Beamline | PDB ID (5JD2) |
| Space group | LCLS (CXI) |
| Cell dimensions | P2₁ |
| $a, b, c$ (Å) | 50.7, 98.4, 53.1 |
| $\alpha, \beta, \gamma$ (°) | 90, 112.7, 90 |
| $R_{split}$ | 0.048 (0.395) |
| $I/\sigma(I)$ | 14.0 (2.7) |
| Completeness (%) | 1.0 (1.0) |
| SFX multiplicity of observations | 1447.6 (1003.3) |
| $CC^*$ | 1.000 (0.930) |
| $CC^{1/2}$ | 0.998 (0.762) |
| $CC^{ano}$ | 0.177 (0.003) |
| Wilson B Factor (Å²) | 29.58 |
| | |
| *Refinement* | |
| Resolution (Å) | 32.51–1.90 (1.97–1.90) |
| No. of reflections | 38,327 (3,817) |
| $R_{work}/R_{free}$ | 0.166/0.199 (0.231/0.253) |
| Ramachandran favoured (%) | 89.3 |
| Ramachandran allowed (%) | 10.5 |
| Ramachandran outliers (%) | 0.2 |
| No. of atoms | |
| Protein | 3,630 |
| Ligand/ion | 64 |
| Water | 265 |
| *B*-factors (Å²) | |
| Protein | 34.0 |
| Ligand/ion | 38.9 |
| Water | 43.5 |
| Root-mean-square deviations | |
| Bond lengths (Å) | 0.006 |
| Bond angles (°) | 1.04 |

*Values in parentheses are for highest-resolution shell.

mother liquor composition of 24% PEG 1,500 and 20% glycerol; however, it has increased background, and the associated fluctuations in the background level added to the challenging nature of the data. The measured $<\Delta F>/<F>$ was ~1% at 2 Å resolution. In addition to the weak $<\Delta F>/<F>$ for the system, no alpha helices were present in the molecule, making the density modification steps challenging and also rendering solutions using automated pipelines ineffective. These challenges necessitated the use of over 300,000 indexed patterns to achieve the successful phasing of the structure. For these reasons, demonstrating that the data for such a challenging system could be used for successful *de novo* structure determination indicates a good possibility for general applicability of Se-SAD *de novo* phasing at LCLS.

Overall, Se-SAD phasing was successfully demonstrated for the SFX data collected at LCLS. This demonstration indicates that an important, standard method for structural biology is now available using X-ray FELs. Se-SAD SFX experiments should allow for challenging and novel macromolecule structures to be determined at ambient temperature using small total quantities of microcrystals, offering new possibilities in crystallography not available with the traditional methods.

## Methods
**LCLS machine parameters.** LCLS was originally designed to operate up to the photon energy of 8.3 keV although photon energies of up to 11.2 keV have recently been delivered almost routinely. The electron beam used to generate the X-rays passes through three linear accelerators and two magnetic bunch compressor chicanes (among other equipment)[33]. In order to achieve ~12.8 keV X-rays, the correlated energy spread of the electron beam after the second linear accelerator was made weaker while the compression factor of the second magnetic bunch compressor chicane was increased. As a result, the accelerator was able to produce X-rays with photon energy up to 13.0 keV. The photon energy distribution was fitted using a Gaussian model with mean photon energy of 12803.40 eV and a standard deviation of 12.13 eV, as shown in Supplementary Fig. 4.

**Experimental design.** The Se-SAD SFX experiments were performed at the CXI instrument of LCLS[34] using 12.8 keV X-rays. The experiments were performed using a serial operation of the serial crystallography set-up[21], in which case the unused X-rays from a primary SFX experiment are refocused by Be compound refractive lenses to a second SFX experiment. Individual X-ray pulses containing ~2 × 10¹¹ photons at 100% transmission with pulse duration between 35 and 40 fs were focused into a spot size of ~3 × 3 µm in both the upstream and downstream experiments.

**Sample preparation.** Streptavidin crystals in the presence of selenium-derivatized biotin molecules were grown as a test sample for Se-SAD[3]. Lyophilized recombinant, high-purity (>98% by SDS–polyacrylamide gel electrophoresis) core-streptavidin protein was purchased from Creative Biomart (Cat# Streptavidin-501). Selenobiotin was purchased from Santa Cruz Biotechnology (Cat # sc-212920). For crystallization screening, selenobiotin was mixed in a 2:1 molar ratio with 25 mg ml⁻¹ streptavidin in 22.5% (volume/volume) 2-methyl-2,4-pentanediol (MPD) and incubated on ice overnight. The mixture was centrifuged at 14,000g for 10 min to separate and discard solid impurities. The final mixture was screened against a library of 3,000 crystallization conditions by combining equal volumes of protein with each crystallization condition in 72 well-format Terasaki microbatch plates. The mixture was covered with 100% paraffin oil to prevent vapor diffusion. This method permitted rapid set-up and observation of protein crystallization with each condition. Se-B SA co-crystals were obtained overnight at room temperature; some of the conditions yielded crystals within a few hours. These conditions were then evaluated visually using a light microscope for quantity and quality of crystals. A limited number of conditions were selected for further optimization and evaluation. Crystals from these conditions were screened for diffraction quality and a complete 1.35-Å data set was collected at beamline 12–2 of the stanford synchrotron radiation lightsource (unpublished results). Diffraction patterns collected at CXI from crystals grown in 24% PEG 1,500 and 20% glycerol routinely extended to a resolution beyond 2 Å (Supplementary Fig. 1).

**Sample introduction.** A coMESH injector was set-up for each sample chamber as described previously[22]. In order for the coMESH geometry to interface with the standard CXI injector rod hardware, the Labsmith Tee was placed outside of vacuum, resulting in a 1.5 m long concentric length of capillaries, as opposed to the shorter 5 cm length reported previously. The inner sample line was 2 m of 100 × 160 µm fused silica capillary from Polymicro (Molex) and connected directly to the custom LCLS sample reservoirs. The sister liquor was the same MPD sister liquor reported in Sierra *et al.*[22]. The concentric capillary was 250 × 360 µm fused silica capillary. High voltage power supplies (Iseg) supplied voltages to the sister liquor less than 5 kV and <1 µA. The flowrate of the sister liquor was adjusted between 1–10 µl min⁻¹ in order to ensure jet stability and quality, sometimes resulting in higher than normal backgrounds.

**Data availability.** Coordinates and structure factors have been deposited in the Protein Data Bank (www.wwpdb.org) under the accession code 5JD2. Data associated with the manuscript can be obtained from the authors in raw or reduced format upon reasonable request.

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

## Acknowledgements

Portions of this research were carried out at the Linac Coherent Light Source (LCLS) at the SLAC National Accelerator Laboratory. LCLS is an Office of Science User Facility operated for the US Department of Energy Office of Science by Stanford University. Use of the Linac Coherent Light Source (LCLS), SLAC National Accelerator Laboratory, is supported by the U.S. Department of Energy, Office of Science, Office of Basic Energy Sciences under Contract No. DE-AC02-76SF00515. H.D., R.G.S., and E.H.D. acknowledge the support of the OBES through the AMOS program within the CSGB and the DOE through the SLAC Laboratory Directed Research and Development Program. Parts of the sample delivery system used at LCLS for this research was funded by the NIH grant P41GM103393, formerly P41RR001209.

## Author contributions

M.S.H., S.W. and S.B. designed and coordinated the project. H.D., R.G.S., E.H.D., R.A., F.A., H.C., P.M., Y.R., L.Z. built the coMESH injectors, prepared and characterized the samples. M.J.H. and S.G. built and installed the hardware to run the CXI instrument at 12.8 keV. M.S.H., C.H.Y., H.D., R.G.S., E.H.D., R.A., F.A., A.L.A., H.C., M.L., U.L., J.K., P.M., Y.R., L.Z. and S.B. conducted the experiments. R.G.S. and E.H.D. operated the coMESH injectors. C.H.Y., T.J.L. and M.S.H. processed the data. M.S.H. reached out to J.M.H. for help analysing the challenging data set. J.M.H. and C.H.Y. phased the data. H.D. refined the structure. M.S.H., C.H.Y., H.D., R.G.S., E.H.D. and S.B. prepared the manuscript with input from all of the authors.

## Additional information

**Accession codes:** Protein Data Bank: coordinates and structure factors of Streptavidin-selenobiotin complex have been deposited under accession code 5JD2.

**Competing financial interests:** The authors declare no competing financial interests.

**Publisher's note**: 

