## [Peer Review File · Nature Communications]

Reviewer #1 (Remarks to the Author):

The authors have addressed all points raised in the first review of the manuscript (when it was under consideration for Nature Methods) and I would strongly recommend to publish the manuscript in Nature Communications.

If I may suggest one minor thing to the authors, then to add the Pande et al. (DOI: 10.1126/science.aad5081) paper as a citation where they cite Barends et al. and Tenboer et al.

Reviewer #2 (Remarks to the Author):

The revised manuscript "Selenium single-wavelength anomalous diffraction de novo phasing using an X-ray Free Electron Laser" by Dr Hunter and colleagues addresses the initial reviewer comments and makes for a compelling read.

The paper has been transferred from Nature Methods and consequently focuses on method development. It continues a series of recent publications demonstrating phasing of protein structures using X-ray free electron lasers (published ones are well referenced). What differentiates the submitted manuscript is the demonstration of Se-SAD, a successful technique widely used at synchrotron sources. This point is worked out well both in terms of the experiment itself and the presentation of the results in the manuscript. I have no doubt about the validity of the made conclusions.

The main question is to what extent the manuscript would be of interest to a wider audience. Even though many were skeptical only a few years ago, I don't think today anybody would doubt that Se-SAD is possible using serial crystallography at an XFEL. If it is a smart thing to do considering the cost of beamtime and the amount of protein necessary is another matter. At least in the presented case of selenobiotinyl-streptavidin the disadvantages seem not to be outweighed by the advantage of outrunning radiation damage. The main application could be in cases where only very small crystals are available but likely these would not diffract well enough for the method to work. Still somebody had to demonstrate and document that it is possible and this certainly warrants publication. Personally I however feel this is better done in a more specialized journal or perhaps in Scientific Reports with its strong focus on technical soundness.

Response to referees

“Selenium single-wavelength anomalous diffraction *de novo* phasing using an X-ray Free Electron Laser”
Hunter *et al.*

Reviewer #1 (Remarks to the Author):

- The authors have addressed all points raised in the first review of the manuscript (when it was under consideration for Nature Methods) and I would strongly recommend to publish the manuscript in Nature Communications.
- If I may suggest one minor thing to the authors, then to add the Pande *et al.* (DOI: 10.1126/science.aad5081) paper as a citation where they cite Barends *et al.* and Tenboer *et al.*
 - We thank the reviewer for the recommendation and have updated the manuscript to reference the Pande *et al.* publication.

Reviewer #2 (Remarks to the Author):

- The revised manuscript “Selenium single-wavelength anomalous diffraction *de novo* phasing using an X-ray Free Electron Laser” by Dr Hunter and colleagues addresses the initial reviewer comments and makes for a compelling read.
- The paper has been transferred from Nature Methods and consequently focuses on method development. It continues a series of recent publications demonstrating phasing of protein structures using X-ray free electron lasers (published ones are well referenced). What differentiates the submitted manuscript is the demonstration of Se-SAD, a successful technique widely used at synchrotron sources. This point is worked out well both in terms of the experiment itself and the presentation of the results in the manuscript. I have no doubt about the validity of the made conclusions.
- The main question is to what extent the manuscript would be of interest to a wider audience. Even though many were skeptical only a few years ago, I don't think today anybody would doubt that Se-SAD is possible using serial crystallography at an XFEL. If it is a smart thing to do considering the cost of beamtime and the amount of protein necessary is another matter. At least in the presented case of selenobiotinyl-streptavidin the disadvantages seem not to be outweighed by the advantage of outrunning radiation damage. The main application could be in cases where only very small crystals are available but likely these would not diffract well enough for the method to work. Still somebody had to demonstrate and document that it is possible and this certainly

warrants publication. Personally I however feel this is better done in a more specialized journal or perhaps in Scientific Reports with its strong focus on technical soundness.

- We thank the reviewer for the comments and discussion. We don't see anything to respond to other than that we believe the work will be of interest to the broader scientific community.